# Xylan Prebiotics and the Gut Microbiome Promote Health and Wellbeing: Potential Novel Roles for Pentosan Polysulfate

**DOI:** 10.3390/ph15091151

**Published:** 2022-09-16

**Authors:** Margaret M. Smith, James Melrose

**Affiliations:** 1Raymond Purves Laboratory of Bone and Joint Research, Kolling Institute of Medical Research, Faculty of Health and Science, University of Sydney at Royal North Shore Hospital, St. Leonards, NSW 2065, Australia; 2Graduate School of Biomedical Engineering, University of New South Wales, Sydney, NSW 2052, Australia; 3Sydney Medical School, Northern Campus, University of Sydney at Royal North Shore Hospital, St. Leonards, NSW 2065, Australia

**Keywords:** xylan, pre-biotics, gut microbiome, pentosan polysulfate, gut symbionts, DMOAD

## Abstract

This narrative review highlights the complexities of the gut microbiome and health-promoting properties of prebiotic xylans metabolized by the gut microbiome. In animal husbandry, prebiotic xylans aid in the maintenance of a healthy gut microbiome. This prevents the colonization of the gut by pathogenic organisms obviating the need for dietary antibiotic supplementation, a practice which has been used to maintain animal productivity but which has led to the emergence of antibiotic resistant bacteria that are passed up the food chain to humans. Seaweed xylan-based animal foodstuffs have been developed to eliminate ruminant green-house gas emissions by gut methanogens in ruminant animals, contributing to atmospheric pollution. Biotransformation of pentosan polysulfate by the gut microbiome converts this semi-synthetic sulfated disease-modifying anti-osteoarthritic heparinoid drug to a prebiotic metabolite that promotes gut health, further extending the therapeutic profile and utility of this therapeutic molecule. Xylans are prominent dietary cereal components of the human diet which travel through the gastrointestinal tract as non-digested dietary fibre since the human genome does not contain xylanolytic enzymes. The gut microbiota however digest xylans as a food source. Xylo-oligosaccharides generated in this digestive process have prebiotic health-promoting properties. Engineered commensal probiotic bacteria also have been developed which have been engineered to produce growth factors and other bioactive factors. A xylan protein induction system controls the secretion of these compounds by the commensal bacteria which can promote gut health or, if these prebiotic compounds are transported by the vagal nervous system, may also regulate the health of linked organ systems via the gut–brain, gut–lung and gut–stomach axes. Dietary xylans are thus emerging therapeutic compounds warranting further study in novel disease prevention protocols.

## 1. Introduction

This study highlights the gut microbiome and how it generates prebiotic metabolites from dietary components with beneficial effects on members of the gut microbiota and linked organ systems in health and disease [1,2,3]. Devani-Davari et al., 2019 [4], state that prebiotics are a group of nutrients that are degraded by gut microbiota, and many additional studies cited in this review show how this leads to the maintenance of the varied cell populations present in the gut microbiome and also has health-related benefits [4]. The term prebiotics should not be confused with the term probiotics. A consensus statement released by the international scientific association for probiotics and prebiotics has recommended a definition for the use of the term probiotics [5] as “live microorganisms that, when administered in adequate amounts, confer a health benefit on the host”. Etymologically the term probiotic is a Greek term meaning “for life”. A prebiotic is “a dietary substrate that is selectively metabolized by the host microbiome to confer a health benefit” [5,6]. Dietary supplements have been suggested to have the ability to combat COVID-19 disease and the gut dysbiosis associated with this condition [7,8] and confectionaries containing phytochemicals with anti SARS CoV-2 activity have even been developed.

Xylans and their microbiome generated metabolites are important prebiotic compounds. The human genome does not encode enzymes with xylanolytic capability, and thus, dietary xylans transit the gastrointestinal tract undegraded, acting as indigestible dietary fibre that promotes throughput of the digested gut contents. However, the intestinal microbiota utilise xylans as a nutrient source and produce xylan metabolites with prebiotic properties that promote gut health. Specific microbiota members have been engineered to produce cytokines, growth factors and other bioactive proteins, and secretion of these compounds can be induced by a xylan induction system [9,10,11]. This is an innovative approach to the treatment of specific diseases and represents a technological revolution through a new therapeutic interface that can be regulated by control of the diet. Thus, therapeutic responses can be effected not only in the gut but also in major linked organ systems such as the liver, lung and brain through the gut–liver [12,13], gut–lung [14,15] and gut–brain [16,17] axes.

## 2. Xylans Are Abundant Plant Carbohydrates

Xylan is the third most abundant naturally occurring carbohydrate biopolymer on Earth after cellulose and chitin [18]. Xylan is a component of the secondary cell walls of dicotyledonous plants, all cell walls of cereals and a major structural carbohydrate of seaweeds [19]. The human genome does not contain xylanolytic enzymes, and as a consequence, dietary xylans transit through the gastrointestinal tract undegraded. They are a major contributor to the fibre content of foods, serving as a bulking agent which promotes the movement of digested food components through the small and large intestine. The gut microbiota produce a range of xylanolytic enzymes that degrade xylans [20] into prebiotic metabolites that regulate the microbiome and significantly contribute to health and well-being [21]. The intestinal epithelium is important in the maintenance of T cells in the gut. Intraepithelial CD8α T cells in close contact with intestinal epithelial cells and the underlying basement membrane aid in the detection of invasive pathogens. T cell survival depends on β_1_ integrin interactions with type IV collagen in the basement membrane. Knock-out of β_1_ integrin expression in CD8α T lymphocytes decreases levels and the migratory properties of intraepithelial T cells in-vivo and the protective responses they provide against pathogenic bacteria. Type IV collagen interactions with β_1_ integrins on intraepithelial T cells are not only important for T cell survival but the provision of T-cell protective properties in mucosal immunity [22].

The gut microbiome contains 10–100 trillion microorganisms [23] that control the digestion of food and regulate the immune [24,25] and central nervous systems [26] and is also linked to other major organ systems like the liver [27] and lung [28]. Gut dysbiosis is associated with long COVID-19 disease [29] and with secondary antibiotic resistant bacterial infections such as *Clostridium difficile* that have emerged in the COVID-19 pandemic, [30] adding a further complication in the treatment of long COVID-19 disease. Some members of the microbiome digest fibre [31] and release short-chain fatty acids [32], which regulate gut health providing gut barrier properties; prevent weight gain [33] and lower cholesterol levels [34] and the incidence of diabetes [35], heart disease [36] and the risk of cancer [37,38,39]. Gut commensal bacteria produce a range of xylanolytic enzymes that allow them to utilise dietary xylans as nutrients [20,40]. Long-chain xylans are one of the most common dietary fibres in the human gastrointestinal tract that promote the growth of *Bifidobacterium pseudocatenulatum* [41]. Xylo-oligosaccharides are prominently generated from xylans by *B*. *pseudocatenulatum*, and these have prebiotic properties that counter gut dysbiosis, [41,42] reducing the inflammatory response in the gut induced by obesity [43]. Methods have been developed to prepare xylans and xylo-oligosaccharides to evaluate their potential health benefits [41,44,45,46,47]. Nutraceutical supplements are being developed to combat COVID-19 disease [48,49] and are also of application in the treatment of critically ill patients [50,51,52,53].

### 2.1. Dietary Xylans

Xylans are complex polysaccharides classified as (i) glucuronoxylans (GXs), (ii) arabinoxylans (AXs) and (iii) glucuronoarabinoxylans (GAXs) based on their constituent monosaccharides [18,54]. Xylans are the second most abundant hemicellulose and represent ~25–35% of the carbohydrate biomass of woody tissues of dicotyledonous plants and lignin rich tissues of monocotyledons, comprising up to 50% by dry weight of grasses and cereal grains. [18,54] Xylans are complex heteropolysaccharides containing a β-(1,4)-glycosidically linked D-xylose backbone, L-arabinose (L-Ara), D-glucuronic acid (D-GlcA), D-GlcA methylated at *O*-4, or acetyl group [18] side chains (Figure 1a,b). These side chains can be further esterified with acetic and ferulic acids [54]. The type and frequency of these side chains and their modifications vary with the tissue source of the xylan and determine whether the xylan has gel-forming properties in situ or acts as a structural carbohydrate [55] (Figure 1a,b). Xylans in woods and cereal stems are acetylated, and they cross-link cellulose fibres, providing high mechanical support to these tissues [56]. Arabinoxylan in the cereal endosperm has water retention properties and maintains the hydration of the seed head [55]. Seaweed xylans occur as 1,3-β-D-xylans; 1,3:1,4-β-D-xylans and 1,4-β-D-linked xylans, and these are assembled into triple-helical microfibrillar structures that have similar supportive properties to the cellulose fibres found in terrestrial plants [57]. Xylans thus have a variable structure and function depending on their tissue of origin. AX is a prominent gel-forming xylan in the endosperm of cereal seed heads and its hydration properties ensure the viability of the embryo in the aleurone layer is maintained [55]. GAX in the secondary plant cell wall of the cereal stem has a mechanically supportive role and is acetylated and substituted with ferulic acid to variable degree [56]. In addition, ferulic acid esters derived from lignin are also found attached to *O*-5 of L-Ara in wood xylans and form a linkage group for the xylan to cellulose fibres. Stem and wood xylans also contain α1–2 or α1–3 linked D-GlcA and 4-*O*-methylated D-GlcA as well as α L-Ara furanose (*α* L-Ara_f_) and *O*-acetyl groups attached to the xylan main chain. Wood xylans are more heavily substituted with acetyl and ferulic acids compared to cereal xylans [58].

### 2.2. Pentosan Polysulfate, a Therapeutic Semi-Synthetic Sulfated Xylan

Pentosan polysulfate (PPS) is a semi-synthetic sulfated xylan produced from beech wood xylan. PPS is heavily sulfated and is referred to as a heparinoid; however, it has a higher charge density than both heparin and heparan sulfate (HS), is less heterogeneous and is a small molecular weight drug (4–6 kDa) (Figure 1c). PPS is a potent disease modifying osteoarthritic drug (DMOAD) [59,60,61], has been used to treat cystitis and painful bowel disorder in humans and has anti-viral properties and potential anti-SARS Cov-2 activity [62,63,64,65]. Approximately one in every ten residues of PPS has a 4-*O*-methylated D-GlcA side chain linked *O*-2 to the xylan backbone, but PPS is devoid of the other xylan modifications mentioned above [66].

## 3. Degradation of Xylans in the Gastrointestinal Tract

The gut microbiota produce xylanolytic enzymes that generate a range of xylan prebiotic metabolites (Figure 1d). Degradation of xylan into xylo oligosaccharides (Figure 1e) and into free xylose requires the combined action of degradative enzymes such as α-L-arabinofuranosidase (EC 3.2.1.55), α-D-glucuronidase (EC 3.2.1.139), acetylxylan esterase (EC 3.1.1.72) and ferulic acid esterases (EC 3.1.1.73), which release the side chains from the xylan backbone. Endo-β-1,4-xylanase (EC 3.2.1.8) acts synergistically with β-xylosidase (EC 3.2.1.37) to degrade the xylan backbone with the former hydrolysing the internal β-(1,4) linkages of the xylan backbone to produce short xylo-oligosaccharides, and β-xylosidase then removes xylose units from the non-reducing termini of these xylo-oligosaccharides (Figure 1e). β-D-xylosidases are a diverse group of enzymes, and several families of xylosidases have been characterised [67]. The L-Ara, D-GlcA, *O*-4 Me D-GlcA side chains of xylans provide solubility to the xylan and some protection to enzymatic depolymerisation of the xylan backbone. Ferulic acid modification of L-Ara side chains provides stabilising attachment points for xylans, connecting them to cellulose fibres and providing mechanical stability in plant tissues (Figure 1a,b).

**Figure 1 pharmaceuticals-15-01151-f001:**
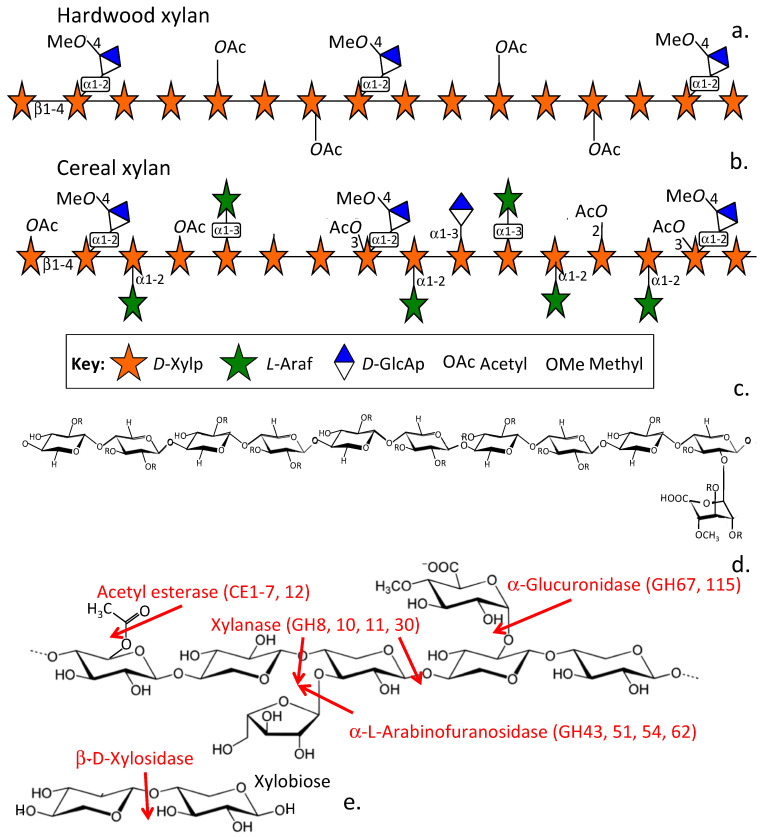
Diagrammatic representations of the diverse features of the structural organisation of hardwood (**a**) and cereal xylans (**b**). Representative features are shown using symbol nomenclature for glycans (SNFG). The main xylan chain is β1–4 glycosidically linked. Xylose (D-Xyl_p_) and Glucuronic acid (D-GlcA_p_) residues occur as pyranose and Arabinose as furanose (L-Ara_f_) ring structures in xylans. Structure of PPS showing its β1–4 linked linear xylan backbone containing a 4-*O*-methyl glucuronate side chains attached *O*-2 to every tenth xylose residue of the main chain. R = sulphate groups (**c**). Xylanolytic enzymes that co-operatively degrade xylans (**d**) into xylo-oligosaccharides and eventually to xylobiose and then to D-xylose (**e**). CAZy [68] enzyme families are indicated in brackets.

Typical xylan side chain structures from beech wood and cereals and their side chains are shown in Figure 2. Xylans are composed of a β1,4-linked D-xylose backbone containing at *O*-2 and/or *O*-3 α-L-Ara_f_ and acetyl groups, and exclusively at *O*-2 with GlcA or 4-*O*-methyl-GlcA (MeGlcA) (Figure 2a–e). Complete enzymatic degradation of xylan requires the co-ordinated action of several enzymes (Figure 1d). The gut microbiome synthesise these xylanolytic enzymes. The xylanases are classified in the CAZy (Carbohydrate-Active enZYmes) database [68] into six different glycosidic hydrolase families, GH5, GH7, GH8, GH1O, GH11 and GH43 (Figure 1d). These differ in specificity, e.g., GH10 xylanases cleave highly substituted xylan chains whereas GH11 only cleaves xylans with at least 3 unsubstituted xylose residues. The CAZy Database (http://www.cazy.org/) is dedicated to the display and analysis of genomic, structural and biochemical information on carbohydrate degradative enzymes [68]. The glucuronidases that hydrolyse the GlcA and MeGlcA decorations on xylan chains are members of the GH67 family [69,70]. Two bacterial phyla, the *Bacteroidetes* and *Firmicutes*, have evolved to degrade complex polysaccharides in the gut such as xylans [71,72,73,74].

Endoxylanases generate prebiotic xylo-oligosaccharides from dietary xylans [20,47]; these promote beneficial symbiont microbes such as *Bifidobacterium* and *Lactobacillus* sp. in the human gut and maintain mucosal health and immune function [75,76,77,78]. Xylo-oligosaccharides also inhibit colonization of the gut by pro-inflammatory bacteria, such as *Salmonella* sp., and promote growth of *Roseburia intestinalis*, an abundant butyrate-producing *Firmicute* which improves gut barrier properties, and plasma lipid levels attenuating pro-inflammatory effects of a high fat diet, decreasing LPS levels in blood and LPS-induced IL-1β and IL-13 [78].

### 3.1. The Xylan Regulon and Production of Xylanolytic Enzymes

The xylan regulon [79] is a cluster of genes in commensal bacteria that encode for a number of xylanolytic enzymes. These genes of the xylan regulon are activated by dietary xylan. Gut commensal bacteria produce a number of endo β-D-xylanase glycohydrolases (GH 30, 10, 11 and 30) which internally cleave β1–4 glycosidically linked xylopyranose residues of the xylan backbone releasing xylo-oligosaccharides. These xylo-oligosaccharides then act as substrates for a family of β-D-xylosidases which release D-xylose monosaccharide units from the non-reducing termini of xylotriose and xylobiose. Other side chain components (depending on xylan source) are released from the xylan backbone by α-L-arabinofuranosidases (GH 43, 51, 54, 62), acetyl esterases (CE 1–7, 12) and α-D-glucuronidases (GH 67,115); GH 1O endoxylanase (Xyn A1) releases arabinoxylobiose, arabinoxylotriose, xylobiose, xylotriose and methylglucuronoxylotriose from glucuronoarabinoxylans (GAXs). Gut commensal bacteria utilize the released monosaccharides as a nutritional source.

### 3.2. Metabolism of GAGs in the Gut Microbiome and the Essential Roles They Play in Gut Homeostasis

Glycosaminoglycans (GAGs) are constant components in the gut and are present as proteoglycans or as free GAG forms. Some members of the gut microbiome produce GAG depolymerizing enzymes that allow GAGs to be used as nutrient sources by the gut microbiome [80]. Gut bacteria produce a range of sulfatases which are used in the degradation of GAGs including heparin and HS, sulfated neurotransmitters such as serotonin and dopamine, and the hormones melatonin, estrone, dehydroepiandrosterone, and thyroxine [81,82,83]. GAGs have essential roles in the regulation of the colonization and proliferation of beneficial symbiont bacterial populations and the prevention of gut colonization by pathogenic bacteria [84]. Digestion of GAGs such as CS, HS, HA and dietary xylans by the microbiome generates short chain fatty acids that improve gut health [84,85,86,87]. GAGs are one of the most important host glycans that are continuously and abundantly present in the intestine through continuous shedding of proteoglycan from the gut epithelium [88]. In murine gut models, CS disaccharides alter the microbiota increasing the prevalence of *Bacteroides acidifaciens* [86], a bacterial strain that inhibits pathogenic colonization in the gut via induction of IgA production [89]. Oral administration of other GAGs also elicits beneficial gut responses promoting growth of *Lactobacillus* bacteria which ensures gut homeostasis [90,91]. Oral HS administered as enoxaparin, a low molecular weight heparin, improves mucosal healing in a murine colitis model [92]. Dietary HS also improves recovery of renal functions in nephrectomized rats [93]. Low molecular weight HA produced by depolymerization of high molecular weight HA has also been shown to reduce membrane permeability associated with colitis [94].

The Bacteroidetes are the second-most abundant bacterial phylum capable of catabolizing a diverse range of polysaccharides, including GAGs, which is attributable to the diverse range of carbohydrate metabolizing enzymes (CAZymes) in their genomes [95,96,97]. *Proteus vulgaris*, a component of a healthy gut microbiome [98], produces two well-characterized chondroitinase enzymes that are not encoded in the human genome; thus, GAGs are of limited nutritive value to mammalian cells but can be utilized by members of the gut microbiome [99,100,101]. Unlike the *Bacteroides*, the ability to degrade GAGs is not widely prevalent in other gut phyla. Salyers et al. [102] showed 154 faecal strains of Firmicutes and *Bifidobacteria* (Actinobacteria) were incapable of metabolizing CS, HA and heparin. Crociani and colleagues [103] also demonstrated that 239 strains of *Bifidobacterium* were incapable of metabolizing heparin, CS, HA and polygalacturonate, and therefore, the abundant Bacteroidetes with GAG-degradative properties have particularly important roles to play in the maintenance of gut homeostasis.

Pathogenic bacteria can colonize gut tissues using GAGs and PGs as a site of entry to infect host cells [104]. Enteric pathogens such as *Toxoplasmosis gondii*, *E*. *coli* O157:H7 [104] and opportunistic pathogenic *Streptococci* [105] can utilise GAGs as entry points to infect intestinal cells. Antibiotics used to treat these gastrointestinal pathogens detrimentally affect beneficial gut bacteria and may cause gut dysbiosis. It is therefore essential that beneficial gut bacteria be maintained as dominant symbionts to ensure that pathogenic bacteria do not obtain a niche to colonise the gut microbiome.

Experiments with radiosulfate-labelled PPS (Elmiron) showed it was metabolised into lower molecular weight forms of lower sulfation, with ~50% of orally-administered PPS absorbed and excreted in the faeces; 11% of radiolabelled PPS was excreted by the kidneys, and ~3% of the PPS was discharged in the urine as intact 4–6 kDa PPS [106]. PPS has efficacy in the treatment of cystitis [107] and urinary tract infections, [108] indicating that sufficient bioactive levels of intact PPS were present in the urinary tract to provide a therapeutic effect in urinogenital infections. However, as already discussed, based on the known GAG degrading capability and xylanolytic enzymes produced by gut microbiota members, PPS would also be expected to be converted to prebiotic xylo-oligosaccharides, with roles in the maintenance of gut health extending the therapeutic profile of PPS.

Sodium alginate is a sulfated polymer commonly used as a food additive in Asian cuisines. An analysis of the human gut microbiota for bacteria capable of degrading alginate into its mannuronic and guluronic acid components also revealed specific members of the *Bacteroides* genus with this capability [109]. Additionally, the sulfate-reducing bacterium *Desulfovibrio piger*, which colonizes the gut of ∼50% of all humans, would also be expected to contribute to alginate degradation [110,111,112,113,114]. Mammalian cells also produce a number of sulfatases that degrade GAGs and sulfated cerebrosides. These include *N*-sulfoglucosamine sulfohydrolase, *N*-acetylglucosamine-6-sulfatase, iduronate 2-sulfatase, *N*-acetylgalactosamine-4-sulfatase, cerebroside sulfatase and *N*-acetylgalactosamine-6-sulfatase [115].

## 4. Xylans Promote a Healthy Human Microbiome and Linked Organ Systems

Digestion of xylans by the microbiome maintains stable health promoting bacterial symbionts in the gut [116]. Carbohydrate epitopes released from dietary polysaccharides by the microbiota educate the human immune system in infancy, aiding in the provision of tolerance to food types and minimising the development of food allergies in later life [117,118,119,120,121]. The microbiota of the gastrointestinal tract have a close symbiotic relationship with the human host with roles in health maintenance, metabolism of indigestible dietary fibre and synthesis of some vitamins and neurotransmitters [122]. The prevalence of beneficial symbiont species in the gut prevents the colonization of the gut by harmful pathogenic cell populations. The human microbiota of the large intestine are dominated by the *Firmicute* and *Bacteroidetes* phyla which represent >90% of the total microbial community [123]. *Roseburia intestinalis*, a butyrate-producing *Firmicute*, is also important in gut health [116]. Metagenomic and transcriptomic studies have identified distinctive signatures in gut microbiota in neural disorders such as autism and bipolar disorders [124,125,126]. Machine learning techniques are also now being applied to the diagnosis of diseases from dynamic changes in the gut microbiome [127]. The gut microbiota produce bioactive fragments of polysaccharides that undergo fermentation in the gut, which educate the immune system in infancy [128], and immunomodulatory responses that subsequently develop in later life [129], such as development of immune tolerance to food groups and prevention of food allergies [130]. This is a rapidly evolving area of intensive investigation using powerful technologies across multiple research disciplines. The FLEXIGUT rationale is an integrated -omics data analysis framework designed to understand exposomic associations with food substances that result in chronic low-grade gut inflammation [131]. FLEXIGUT aims to characterize human life-course environmental exposures to specific substances and how they impact on gut inflammation and resultant instructive immune responses. Available evidence shows gut metabolites impact on cell populations in vivo affecting cellular responses to fat storage, hypolipidemia, hypoglycaemia, appetite and disease processes [131].

A human study comparing the impact of diet versus drugs on the control of cellular metabolism found that diet had as strong an impact as drugs on many cellular processes and diseases such as obesity, diabetes, heart disease and neurological diseases [126,132,133,134,135,136]. Diet is a powerful medicine, involving nutrient-signalling pathways effecting the gut microbiome [137]. The formation of a healthy microbiome in early childhood is important for the establishment and maintenance of human health in later life. The full impact of the gut microbiome on the attainment of tolerance to certain foods and the neurological pathways that train innate immune responses and how these impact on allergic and autoimmune disorders however is incompletely understood [117,118,119,120,121].

## 5. Manipulation of the Microbiome Increases Farm Animal Productivity

### 5.1. Use of Recombinant Xylanase as a Food Additive for Monogastric Farm Animals Improves Feedstuff Utilization and Animal Productivity

Australian ruminants have an extensive microbiome containing bacteria which produce a vast repertoire of digestive enzymes that can efficiently degrade native Australian grasses that have a high lignin and xylan content. A recombinant β-D-xylanase (Ronozyme^®^ wx2000) has been developed as a feedstuff additive to unlock the nutritive properties of xylans in foodstuffs for monogastric farmland animals to improve the nutrient yield of corns and cereals commonly used as foodstuffs for poultry and pigs [138,139,140,141]. This results in increased performance and optimal growth of these animals, improves food efficiency, and reduces production of waste products and emissions. An increased gut health in terms of elimination of harmful bacterial populations in the gut microbiome also translates into healthier more productive animals.

### 5.2. Elimination of Antibiotic Supplementation in Animal Foodstuffs

The use of prebiotic-generated phytonutrients has been shown to represent an alternative to the use of antibiotics to maintain animal productivity [142]. Prebiotic-generated phytochemicals have been proposed as an ideal alternative to the use of antibiotics in the poultry industry [143]. Dietary xylo-oligosaccharide supplements and xylanase additives increase body weight gain and beneficial gut fermentation metabolites in broiler chickens fed on corn-based diets leading to improved feed utilization and gut health and a reduced need for the use of dietary antibiotics [144]. Ferulic acid (FA), a major phenolic metabolite of plant hemicelluloses, released upon degradation of these carbohydrates by gut microbiota, protects against oxidative stress and inflammation in the gut, promoting animal health and more efficient feedstuff utilization. FA also decreases serum interleukin (IL)-1β, IL-2, IL-6 and tumour necrosis factor (TNF)-α levels and their impact on gut inflammation, which inhibit animal growth and commercial productivity. FA also increases the growth of beneficial symbionts such as the gut *Firmicutes* and *Bacteroidetes*. The alleviation of inflammation and oxidative stress by FA and its ability to maintain a healthy gut microbiome results in the superior performance of farm animals on feedstuffs they would normally digest poorly without help from the gut microbiota [145]. Metagenomics has been used to examine the chicken microbiome genome to better understand how it impacts on health and weight gain efficiency. As in humans, the microbial populations in chicken gut modulate metabolic functions, feed efficiency and health due to the chicken’s ability to digest carbohydrates. This not only produces growth promoting energy-rich metabolites but also aids in the homeostasis of a healthy microbiome [146]. The bovine rumen microbial community also harbours potent carbohydrate-digestive enzyme systems that are particularly efficient in the digestion of plant biomass [147]. The intestinal microbiota are also recognized to have important roles to play in healthy piglet development through effects on intestinal maturation, education of the immune system and development of tolerance to new foodstuffs improving piglet health and growth performance [148].

### 5.3. Manipulation of the Gut Microbiome to Reduce Methane Emissions and Atmospheric Pollution by Ruminant Animals

Agriculture is the largest contributor to global methane emissions, and ruminants (cattle, sheep and goats) are the dominant animal contributors. Carbon emissions from agriculture contribute around 13% of the carbon footprint of Australia, and of these, 43% are from methane produced by ruminant animals. Methane emissions are estimated to increase 30% by 2050 if current agricultural practices are not amended, yet few countries have set methane reduction targets. Rumen microbiota (methanogens) produce methane from the fermentation of polysaccharides that is then belched out into the atmosphere. Seaweed-based animal feedstuffs are rich in prebiotic hemicelluloses, including xylans, which constitute 25–30% of their carbohydrate biomass. This seaweed-based feedstuff promotes the growth of beneficial symbiont gut bacterial populations and eliminates colonization of the gut microbiome by pathogenic cell populations [149,150,151] or methanogens that generate methane [152,153]. A collaboration between Meat & Livestock Australia and James Cook University has developed a cost-effective seaweed foodstuff called FutureFeed [154]. This utilizes a native Australian seaweed in a nutritious animal foodstuff product that significantly reduces methane emissions from ruminant livestock but concomitantly increases livestock productivity and the quality of meat production. Methane is the second most abundant greenhouse gas after CO_2_ and is present at ~20% of the levels reported for atmospheric CO_2_. Methane however is 25–80 times more effective than CO_2_ at trapping heat in the atmosphere. A significant reduction in methane emissions from ruminant livestock can thus make a significant positive contribution towards improvement in global atmospherics contributing to a slowing down of global climate change with no loss in agricultural productivity.

### 5.4. The Health Promoting Properties of Xylo-Oligosaccharide Prebiotics in the Treatment of Human Gut Dysbiosis

Xylo-oligosaccharides are an emerging prebiotic which promote gut health [155], countering the gut inflammation which occurs after antibiotic treatment and the recovery of beneficial *Lactobacillus*, *Bifidobacterium*, *Firmicutes* cell populations in the gut microbiome, previously depleted by antibiotic treatment [76,156]. Re-establishing these gut bacterial populations also prevents colonization of the gut by pathogenic bacterial cell populations [156] and has been found to have beneficial properties associated with the treatment of neurological disorders of cognitive decline through the gut–brain axis [157,158].

### 5.5. The Use of Engineered Commensal Bacteria to Deliver Bioactive Therapeutic Compounds Controlled by a Xylan Protein Induction System

Engineered commensal food-grade bacteria have been developed for the delivery of anti-inflammatory cytokines and other biologically active molecules in the gut. *Lactococcus lactis* has been engineered to co-express tetanus toxin, murine IL-2 or IL-6, a novel vaccination route [159]. The *L*. lactis system secretes active IL-10 as a delivery system in inflammatory bowel disease models and in patients with Crohn’s disease [160]. *L*. *lactis* is not a colonizing bacterium and, combined with limited bioactive growth factor half-lives, repeat therapeutic dosing strategies are required for the treatment of inflammatory bowel disease [161,162]. An alternative strategy has been developed using engineered *Bacteroides ovatus*, a commensal anaerobic Gram-negative bacterium of the colon. Growth factor secretion is regulated by a xylan induction system [163,164,165,166,167]. Keratinocyte growth factor-2 (KGF-2) and TGF-β are epithelial growth factors with immunoregulatory properties of therapeutic value in the treatment of inflammatory bowel disease. The use of engineered *B*. *ovatus* for focal delivery of KGF-2 and TGF-β has considerable potential in the treatment of inflammatory bowel disease. *Bacteroides* spp. are prominent commensal anaerobes found in the mucin layer coating the colonic mucosa and thus ideally placed for therapeutic protein delivery to the injured epithelium. The ability of *B*. ovatus to utilise xylan as its sole carbon source contributes to its predominance as a representative of the colon microbiota. Desulfated PPS is also capable of acting as a xylan for induction of selected proteins by the gut microbiota. This represents a novel level of therapeutic sophistication for PPS. This aspect of the prebiotic therapeutic application of PPS warrants further evaluation.

## 6. Conclusions

Microbiome enzyme systems of the human gut are capable of degrading dietary xylans generating prebiotic metabolites. PPS, a semi-synthetic xylan heparinoid is an efficacious DMOAD that has also been used in the treatment of cystitis and painful bladder disorders and as an anti-arthritic. Biotransformation of PPS by the gut microbiota converts it to a health promoting xylan prebiotic extending its therapeutic profile. Xylan derived oligosaccharides produced by the gut microbiome are emerging agents in the promotion of human health. These have been examined in the treatment of a number of chronic conditions affecting the brain, liver and lung tissues through the gut–brain, gut–liver and gut–lung axes which are served by the vagus nerve which provides a regulatory two-way communication system for the autonomic and parasympathetic nervous systems. These novel prebiotic procedures display significant beneficial therapeutic potential warranting further experimental examination in what represents a new therapeutic frontier.

## Figures and Tables

**Figure 2 pharmaceuticals-15-01151-f002:**
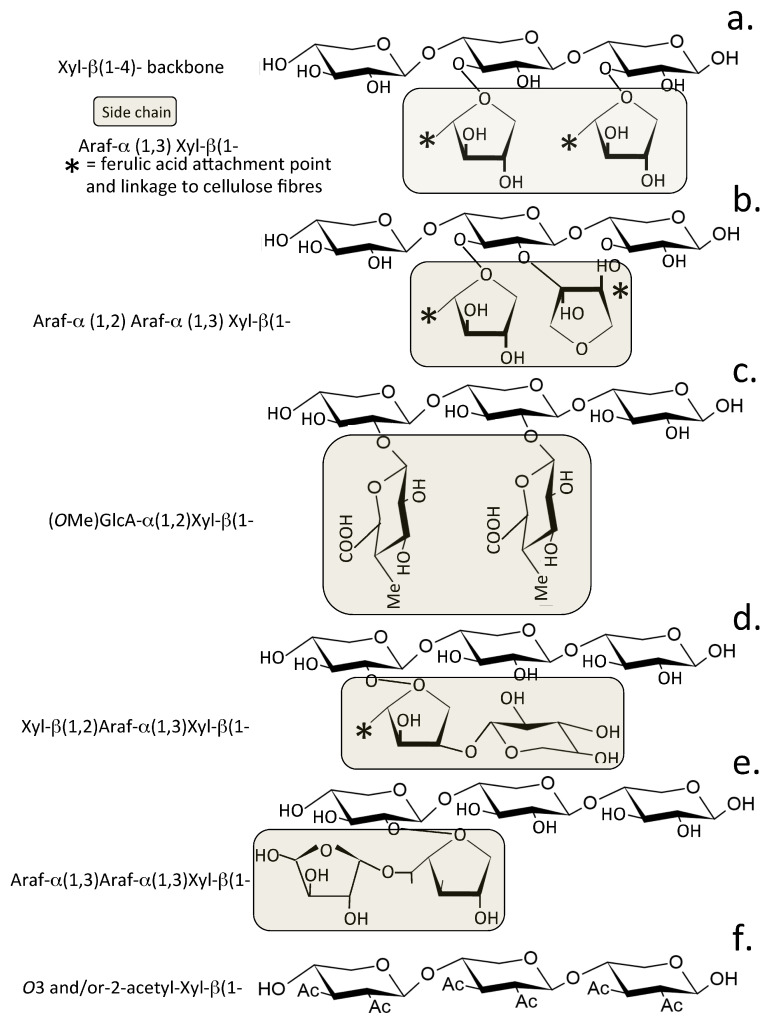
Side chain components found on xylans are attachment sites for feruloyl groups on α1,3-linked Ara_f_, these are labelled with asterisks; Ac, *O*-acetyl groups; Me, *O*-methyl groups. The structures shown depict side chain substitution on a typical arabinoxylan (**a**), di-substituted L-Arabinose side chains in an arabinoxylan (**b**), Glucuronoxylan (**c**), Glucuronoarabinoxylan (**d**) alternative di-Arabinose substitution on an arabinoxlylan to that depicted in b (**e**), *O*-2 and *O*-3 acetylated xylan backbone (**f**).

## Data Availability

Not applicable.

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
