# Peer review of "Xylan Prebiotics and the Gut Microbiome Promote Health and Wellbeing: Potential Novel Roles for Pentosan Polysulfate"

_pharmaceuticals, 2022, doi:10.3390/ph15091151_

Round 1

Reviewer 1 Report

It is a good review with adequate conclusions, however, the words prebiotic and probiotic are sometimes erroneously used.

p1 L24. Use "probiotic-generated molecule" instead of "probiotic molecule"

p1 L26-27. In the figure use "Elimination" instead of "Elemenation".

p1 L33. A probiotic is NOT "a substance", it is a microorganism.

p1 L35, 39. A probiotic is NOT "a compound". The phrase "probiotic-generated compound" is better.

p1 L42. The text says "metabolites with probiotic properties". Did the authors mean "prebiotic properties"?

p4 L116. Again, use "probiotic-generated metabolites" instead of "probiotic metabolites".

p5 L140. Use "energy" instead of "calories"

p6 L148. Delete "probiotic and"

p6 L167. Please define GAG.

p6 L174. Please define CS and HA.

p6 L186. Please define CAZymes.

p7 L197. The correct word is "Toxoplasma", not "Toxoplasmosis"

p7 L210. Again, use "probiotic-generated" instead of "probiotic".

p8 L261 and p9 L289. Use "probiotic-derived" instead of "probiotic".

p9 L302. Use "emerging prebiotics" instead of "an emerging probiotic".

p10 L328. Use "prebiotic" instead of "probiotic".

Author Response

Xylan review Reviewer comments Author responses are in red.

Reviewer 1 Pharmaceuticals 1797214

Top of Form

Comments and Suggestions for Authors

The manuscript entitled "Probiotics, a New Therapeutic Frontier : Animal Studies Show Dietary Xylans Improve Gut Health, Animal Productivity, Obviate Antibiotic Supplementation and Eliminate Atmospheric Pollution By Methanogens." is an interesting story which consist of a bunch of reported informations. 

I would like to suggest few suggestions before a final decision by Chief Editor.

  1. The title of the review is too long and non-impressive, It will be more good if authors shorten it. 
  2. The authors has started the introduction with the sentence "The aim of this......" looks not attractive, therefore try to start with background of the study. This introductory comment has been changed in the revised manuscript.
  3. The authors has associated the role of Xylans and their microbiome generated metabolites with different diseases, very good; it is recommended to add clinical data, especially in case of major diseases e.g. COVID-19, SARS CoV-2, cancer, asthma etc.

No clinical data is available yet just data from preliminary studies advocating the probiotic route as a new way to treat COVID-19 and other diseases. This is the point we were trying to make in the title naming this as a potential new therapeutic frontier. A segment has been added to the revised manuscript to reflect this.

  1. Nayebi A, Navashenaq JG, Soleimani D, Nachvak SM. Probiotic supplementation: A prospective approach in the treatment of COVID-19. Nutr Health. 2022 Jun;28(2):163-175.
  2. Kaushal A, Noor R. Association of Gut Microbiota with Inflammatory Bowel Disease and COVID-19 Severity: A Possible Outcome of the Altered Immune Response. Curr Microbiol. 2022 May 5;79(6):184.
  3. Zhang L, Xu Z, Mak JWY, Chow KM, Lui G, Li TCM, Wong CK, Chan PKS, Ching JYL, Fujiwara Y, Chan FKL, Ng SC. Gut microbiota-derived synbiotic formula (SIM01) as a novel adjuvant therapy for COVID-19: An open-label pilot study. J Gastroenterol Hepatol. 2022 May;37(5):823-831.
  4. Gharajeh NH, Pourjafar H, Derakhshanian H, Mohammadi H, Barzegari A, Eslami S. Gut Microbiota Might Act as a Potential Therapeutic Pathway in COVID-19. Curr Pharm Biotechnol. 2022 Apr 4. doi: 10.2174/1389201023666220404183859.
  5. Liu Y, Kuang D, Li D, Yang J, Yan J, Xia Y, Zhang F, Cao H. Roles of the gut microbiota in severe SARS-CoV-2 infection. Cytokine Growth Factor Rev. 2022 Feb;63:98-107.
  6. Roy K, Agarwal S, Banerjee R, Paul MK, Purbey PK. COVID-19 and gut immunomodulation. World J Gastroenterol. 2021 Dec 14;27(46):7925-7942.
  7. Zhang L, Han H, Li X, Chen C, Xie X, Su G, Ye S, Wang C, He Q, Wang F, Huang F, Wang Z, Wu J, Lai T. Probiotics use is associated with improved clinical outcomes among hospitalized patients with COVID-19. Therap Adv Gastroenterol. 2021 Aug 4;14:17562848211035670.
  8. Hung YP, Lee CC, Lee JC, Tsai PJ, Ko WC. Gut Dysbiosis during COVID-19 and Potential Effect of Probiotics. Microorganisms. 2021 Jul 28;9(8):1605.
  9. de Oliveira GLV, Oliveira CNS, Pinzan CF, de Salis LVV, Cardoso CRB. Microbiota Modulation of the Gut-Lung Axis in COVID-19. Front Immunol. 2021 Feb 24;12:635471.

Administration of probiotic compounds to SARS CoV-2 infected patients counters detrimental changes in the gastrointestinal and respiratory tracts seen in COVID-19 disease [1]. Probiotics suppress severe immune responses and prevent the cytokine storm in COVID-19 disease inhibiting the development of pathologic inflammatory conditions via modulation of immune responses [6] lowering the impact of this disease [2]. A proof of concept study and others which addressed the treatment of gut dysbiosis using a probiotic approach have yielded encouraging results with a significant reduction in pro-inflammatory immune markers, and restoration of gut dysbiosis in hospitalised COVID-19 patients [3]. This therapeutic approach has been advocated as a potential therapeutic pathway in the treatment of COVID-19 disease [4, 5, 7-9].

  1. The Caption of Fig 2 looks incomplete. The complete legend should read :-

Figure 3. Side chain components found on xylans. Attachment sites for feruloyl groups on a1,3-linked Araf are labelled with asterisks; Ac, -O-acetyl groups; Me, O-methyl groups. This has been amended in the revised manuscript

  1. Through out the manuscript the authors used to fill huge literature, either required or not; e.g. at page 5 line 145 "Two bacterial phyla, the Bacteroidetes and Firmicutes, have evolved to degrade complex polysaccharides in the gut such as xylans [101-104]" There is no further information that how they play role in health or body mechanisms. Therefor, the authors are advised to pass the manuscript through effective literature filter. 

This offending segment has been removed from the revised manuscript.

  1. The headings are also long enough; if possible, try to shrink them.  We have shortened the title and headings where we can in the revised manuscript.

The title of the manuscript has been shortened to Probiotics, a New Therapeutic Frontier for the treatment of human disease.

It is a good review with adequate conclusions, however, the words prebiotic and probiotic are sometimes erroneously used.

p1 L24. Use "probiotic-generated molecule" instead of "probiotic molecule" done

p1 L26-27. In the figure use "Elimination" instead of "Elemenation". -changed

p1 L33. A probiotic is NOT "a substance", it is a microorganism. changed

p1 L35, 39. A probiotic is NOT "a compound". The phrase "probiotic-generated compound" is better. The text has been duly amended

p1 L42. The text says "metabolites with probiotic properties". Did the authors mean "prebiotic properties"? yes pre-biotics text changed.

p4 L116. Again, use "probiotic-generated metabolites" instead of "probiotic metabolites". Text changed

p5 L140. Use "energy" instead of "calories" text now changed.

p6 L148. Delete "probiotic and" text duly corrected

p6 L167. Please define GAG. done

p6 L174. Please define CS and HA. Text has been amended

p6 L186. Please define CAZymes. Now defined in text

p7 L197. The correct word is "Toxoplasma", not "Toxoplasmosis" term now corrected in revision

p7 L210. Again, use "probiotic-generated" instead of "probiotic". changed

p8 L261 and p9 L289. Use "probiotic-derived" instead of "probiotic". changed

p9 L302. Use "emerging prebiotics" instead of "an emerging probiotic". changed

p10 L328. Use "prebiotic" instead of "probiotic". done

Reviewer 2 Report

The manuscript entitled "Probiotics, a New Therapeutic Frontier : Animal Studies Show Dietary Xylans Improve Gut Health, Animal Productivity, Obviate Antibiotic Supplementation and Eliminate Atmospheric Pollution By Methanogens." is an interesting story which consist of a bunch of reported informations. 

I would like to suggest few suggestions before a final decision by Chief Editor.

1. The title of the review is too long and non-impressive, It will be more good if authors shorten it. 

2. The authors has started the introduction with the sentence "The aim of this......" looks not attractive, therefore try to start with background of the study.

3. The authors has associated the role of Xylans and their microbiome generated metabolites with different diseases, very good; it is recommended to add clinical data, especially in case of major diseases e.g. COVID-19, SARS CoV-2, cancer, asthma etc.

4. The Caption of Fig 2 looks incomplete

5. Through out the manuscript the authors used to fill huge literature, either required or not; e.g. at page 5 line 145 "Two bacterial phyla, the Bacteroidetes and Firmicutes, have evolved to degrade complex polysaccharides in the gut such as xylans [101-104]" There is no further information that how they play role in health or body mechanisms. Therefor, the authors are advised to pass the manuscript through effective literature filter. 

6. The headings are also long enough; if possible, try to shrink them. 

Regards

Author Response

Reviewer 2 Pharmaceuticals 1797214

Top of Form

Comments and Suggestions for Authors

The manuscript entitled "Probiotics, a New Therapeutic Frontier : Animal Studies Show Dietary Xylans Improve Gut Health, Animal Productivity, Obviate Antibiotic Supplementation and Eliminate Atmospheric Pollution By Methanogens." is an interesting story which consist of a bunch of reported informations. 

I would like to suggest few suggestions before a final decision by Chief Editor.

  1. The title of the review is too long and non-impressive, It will be more good if authors shorten it. 
  2. The authors has started the introduction with the sentence "The aim of this......" looks not attractive, therefore try to start with background of the study. This introductory comment has been changed in the revised manuscript.
  3. The authors has associated the role of Xylans and their microbiome generated metabolites with different diseases, very good; it is recommended to add clinical data, especially in case of major diseases e.g. COVID-19, SARS CoV-2, cancer, asthma etc.

No clinical data is available yet just data from preliminary studies advocating the probiotic route as a new way to treat COVID-19 and other diseases. This is the point we were trying to make in the title naming this as a potential new therapeutic frontier. A segment has been added to the revised manuscript to reflect this.

  1. Nayebi A, Navashenaq JG, Soleimani D, Nachvak SM. Probiotic supplementation: A prospective approach in the treatment of COVID-19. Nutr Health. 2022 Jun;28(2):163-175.
  2. Kaushal A, Noor R. Association of Gut Microbiota with Inflammatory Bowel Disease and COVID-19 Severity: A Possible Outcome of the Altered Immune Response. Curr Microbiol. 2022 May 5;79(6):184.
  3. Zhang L, Xu Z, Mak JWY, Chow KM, Lui G, Li TCM, Wong CK, Chan PKS, Ching JYL, Fujiwara Y, Chan FKL, Ng SC. Gut microbiota-derived synbiotic formula (SIM01) as a novel adjuvant therapy for COVID-19: An open-label pilot study. J Gastroenterol Hepatol. 2022 May;37(5):823-831.
  4. Gharajeh NH, Pourjafar H, Derakhshanian H, Mohammadi H, Barzegari A, Eslami S. Gut Microbiota Might Act as a Potential Therapeutic Pathway in COVID-19. Curr Pharm Biotechnol. 2022 Apr 4. doi: 10.2174/1389201023666220404183859.
  5. Liu Y, Kuang D, Li D, Yang J, Yan J, Xia Y, Zhang F, Cao H. Roles of the gut microbiota in severe SARS-CoV-2 infection. Cytokine Growth Factor Rev. 2022 Feb;63:98-107.
  6. Roy K, Agarwal S, Banerjee R, Paul MK, Purbey PK. COVID-19 and gut immunomodulation. World J Gastroenterol. 2021 Dec 14;27(46):7925-7942.
  7. Zhang L, Han H, Li X, Chen C, Xie X, Su G, Ye S, Wang C, He Q, Wang F, Huang F, Wang Z, Wu J, Lai T. Probiotics use is associated with improved clinical outcomes among hospitalized patients with COVID-19. Therap Adv Gastroenterol. 2021 Aug 4;14:17562848211035670.
  8. Hung YP, Lee CC, Lee JC, Tsai PJ, Ko WC. Gut Dysbiosis during COVID-19 and Potential Effect of Probiotics. Microorganisms. 2021 Jul 28;9(8):1605.
  9. de Oliveira GLV, Oliveira CNS, Pinzan CF, de Salis LVV, Cardoso CRB. Microbiota Modulation of the Gut-Lung Axis in COVID-19. Front Immunol. 2021 Feb 24;12:635471.

Administration of probiotic compounds to SARS CoV-2 infected patients counters detrimental changes in the gastrointestinal and respiratory tracts seen in COVID-19 disease [1]. Probiotics suppress severe immune responses and prevent the cytokine storm in COVID-19 disease inhibiting the development of pathologic inflammatory conditions via modulation of immune responses [6] lowering the impact of this disease [2]. A proof of concept study and others which addressed the treatment of gut dysbiosis using a probiotic approach have yielded encouraging results with a significant reduction in pro-inflammatory immune markers, and restoration of gut dysbiosis in hospitalised COVID-19 patients [3]. This therapeutic approach has been advocated as a potential therapeutic pathway in the treatment of COVID-19 disease [4, 5, 7-9].

  1. The Caption of Fig 2 looks incomplete. The complete legend should read :-

Figure 3. Side chain components found on xylans. Attachment sites for feruloyl groups on a1,3-linked Araf are labelled with asterisks; Ac, -O-acetyl groups; Me, O-methyl groups. This has been amended in the revised manuscript

  1. Through out the manuscript the authors used to fill huge literature, either required or not; e.g. at page 5 line 145 "Two bacterial phyla, the Bacteroidetes and Firmicutes, have evolved to degrade complex polysaccharides in the gut such as xylans [101-104]" There is no further information that how they play role in health or body mechanisms. Therefor, the authors are advised to pass the manuscript through effective literature filter. 

This offending segment has been removed from the revised manuscript.

  1. The headings are also long enough; if possible, try to shrink them.  We have shortened the title and headings where we can in the revised manuscript.

The title of the manuscript has been shortened to Probiotics, a New Therapeutic Frontier for the treatment of human disease.

Reviewer 3 Report

Throughout the manuscript, authors are confused in defining probiotic and prebiotic. and failed to describe whether xylan is a probiotic or prebiotic. The authors were failed to describe the context of the manuscript.

Author Response

Reviewer 3 Pharmaceuticals 1797214

Comments and Suggestions for Authors

Throughout the manuscript, authors are confused in defining probiotic and prebiotic. and failed to describe whether xylan is a probiotic or prebiotic. The authors were failed to describe the context of the manuscript.

Reviewer 1 also raised the issue of the use of pro- and pre-biotics and we have amended our use of these terms following his recommendations throughout the revised manuscript. Xylan is a pre-biotic as we have indicated in the revised manuscript.

The title of the manuscript has been shortened to Probiotics, a New Therapeutic Frontier for the treatment of human disease.

Round 2

Reviewer 2 Report

The authors have responded the comments accordingly. However, need to fix minor spellings and grammar.

Author Response

Probiotics MS Reviewer 2

Comments and Suggestions for Authors

The authors have responded the comments accordingly. However, need to fix minor spellings and grammar.

Author response

The spelling and grammer have been addressed in our revised manuscript R2.

Reviewer 3 Report

Although the article is improved to some extent. I still have following observations and comments

1. Line 2. Title is still very generalized. It gives the impression of therapeutic use of probiotics for the treatment of human disease which is very poorly supported by the data presented. 

It still lacks clarity about what authors are trying to establish, if it is about the role of probiotics in prevention and control of different diseases that is already reviewed by different groups. Is it about role of Xylan (a prebiotic) in prevention of different disease, If so then Xylanolytic enzymes are also present in indegenous microbiota therefore, if the authors are trying to establish a beneficial effects of Probiotics having xylanolytic capacity, they should focus on those kind of probiotics. And if the authors are trying to establish the prebiotic role of xylan, then they need to take into account that xylanolytic bacteria can be the microbiota which can be strengthened by addition of such probiotics as well. 

Line 35. Probiotics are live microbes. Definition is not consistent here

Line 38.  replace probiotic properties with prebiotic properties. 

Line 50-52. Do probiotics produce cytokines or induce the cytokine production by host cells. 

LINE 66-76. This sentence is not understandable. Probiotic metabolites should be replaced with metabolites.

Line 73. Dysbiosis is associated with many other conditions as well.   

Line 91-93. Referring to the study Neyebi et al. their conclusions are exaggerated here. The said study was on a review probiotics role in immune modulation by probiotics and they suggested a possible role in COVID-19 as well.  

Overall, probiotic effects are strain specific and are dependent on the genetics of specific strains with specific mechanisms.  There are multiple mechanisms of probiotics to alleviate the dysbiosis caused by different conditions and antibiotics use. 

Although this review article presents some interesting  studies, it needs redesigning focused on either i) role of probiotics in therapeutics, or ii) role of Xylanolytic probiotics in therapeutics or iii) role of Xylans in enhancing  gut microbial diversity.

In current form it lacks cohesion and cannot be considered for publication. 

Author Response

Reviewer 3

Reviewer comment

  1. Line 2. Title is still very generalized. It gives the impression of therapeutic use of probiotics for the treatment of human disease which is very poorly supported by the data presented.  It still lacks clarity about what authors are trying to establish, if it is about the role of probiotics in prevention and control of different diseases that is already reviewed by different groups. Is it about role of Xylan (a prebiotic) in prevention of different disease, If so then Xylanolytic enzymes are also present in indegenous microbiota therefore, if the authors are trying to establish a beneficial effects of Probiotics having xylanolytic capacity, they should focus on those kind of probiotics. And if the authors are trying to establish the prebiotic role of xylan, then they need to take into account that xylanolytic bacteria can be the microbiota which can be strengthened by addition of such probiotics as well. 

Author response:  The shortened title was in response to one of the other reviewers.  We have now provided an extended title in response to your request which better reflects the scope of our study.

Xylan Prebiotics Promote Gut Health, Obviate Antibiotic Use In Animal Husbandry, Eliminate Ruminant Methane Emissions and have roles in Microbiome-Mediated Human Health

We are well aware of the involvement of microbiome xylanolytic enzyme systems in the gut I would refer you to our comments

Gut commensal bacteria produce a range of xylanolytic enzymes that allows them to utilise dietary xylans as nutrients  Long-chain xylans are one of the most common dietary fibres in the human gastrointestinal tract that promote the growth of Bifidobacterium pseudocatenulatum. Xylo-oligosaccharides are prominently generated from xylans by B. pseudocatenulatum and their prebiotic and probiotic properties counter gut dysbiosis reducing the inflammatory response in the gut induced by obesity.

and

The gut microbiota produce xylanolytic enzymes that generate a range of xylan probiotic generated metabolites (Fig 1d).  Degradation of xylan into xylo oligosaccharides (Fig 1e) and into free xylose requires the combined action of degradative enzymes such as α-L-arabinofuranosidase (EC 3.2.1.55), α-D-glucuronidase (EC 3.2.1.139), acetylxylan esterase (EC 3.1.1.72), and ferulic acid esterases (EC 3.1.1.73), which release the side chains from the xylan backbone. Endo-β-1,4-xylanase (EC 3.2.1.8), acts synergistically with β-xylosidase (EC 3.2.1.37) to degrade the xylan backbone with the former hydrolysing the internal β-(1,4) linkages of the xylan backbone to produce short xylo-oligosaccharides, β-xylosidase then removes xylose units from the non-reducing termini of these xylo-oligosaccharides (Fig 1e). 

I would also refer you to Figure 1(d) which identifies the CAZy enzyme families involved in xylan degradation.

Reviewer comment

Line 35. Probiotics are live microbes. Definition is not consistent here.

Author response

This comment has been rectified in the revised manuscript.

Reviewer comment

Line 38.  replace probiotic properties with prebiotic properties. 

Author response

 Erroneous comment is replaced in the revised manuscript.

Reviewer comment

Line 50-52. Do probiotics produce cytokines or induce the cytokine production by host cells. 

Author response

Specific microbiota members have been engineered to produce cytokines, growth factors and other bioactive proteins and secretion of these compounds can be induced by a xylan protein induction system.

Reviewer comment

LINE 66-76. This sentence is not understandable. Probiotic metabolites should be replaced with metabolites.

Author response

 Comment replaced in the revised manuscript.

Reviewer comment

Line 73. Dysbiosis is associated with many other conditions as well.   

Author response

We have revised the text.

Reviewer comment

Line 91-93. Referring to the study Neyebi et al. their conclusions are exaggerated here. The said study was on a review probiotics role in immune modulation by probiotics and they suggested a possible role in COVID-19 as well.  

 Author response

 This comment has been replaced in the revised manuscript. 

Reviewer comment

Overall, probiotic effects are strain specific and are dependent on the genetics of specific strains with specific mechanisms.  There are multiple mechanisms of probiotics to alleviate the dysbiosis caused by different conditions and antibiotics use. 

Although this review article presents some interesting  studies, it needs redesigning focused on either i) role of probiotics in therapeutics, or ii) role of Xylanolytic probiotics in therapeutics or iii) role of Xylans in enhancing  gut microbial diversity.

In current form it lacks cohesion and cannot be considered for publication. 

Author response

We have produced a more focussed revised manuscript taking your points on board.  The definition of probiotics  we had been using was outdated and consequently the majority of our comments on probiotics were therefore incorrect and inappropriate.  We have addressed this in the revised manuscript.  The focus of the revised manuscript is on xylans as prospective prebiotics.